# Bioactivity of a Novel Polycaprolactone-Hydroxyapatite Scaffold Used as a Carrier of Low Dose BMP-2: An In Vitro Study

**DOI:** 10.3390/polym13030466

**Published:** 2021-02-01

**Authors:** Pawornwan Rittipakorn, Nuttawut Thuaksuban, Katanchalee Mai-ngam, Satrawut Charoenla, Warobon Noppakunmongkolchai

**Affiliations:** 1Department of Oral and Maxillofacial Surgery, Faculty of Dentistry, Prince of Songkla University, Hatyai, Songkhla 90110, Thailand; paworn.r@gmail.com; 2Ministry of Higher Education Science Research and Innovation (MHESRI), Ratchathewi, Bangkok 10400, Thailand; katanchalee.mai@nstda.or.th (K.M.-n.); satrawut.cha@nstda.or.th (S.C.); warobon.nop@nstda.or.th (W.N.)

**Keywords:** polycaprolactone, hydroxyapatite, BMP-2, scaffold

## Abstract

Scaffolds of polycaprolactone-30% hydroxyapatite (PCL-30% HA) were fabricated using melt stretching and multilayer deposition (MSMD), and the in vitro response of osteoblasts to the scaffolds was assessed. In group A, the scaffolds were immersed in 10 µg/mL bone morphogenetic protein-2 (BMP-2) solution prior to being seeded with osteoblasts, and they were cultured in the medium without BMP-2. In group B, the cell-scaffold constructs without BMP-2 were cultured in medium containing 10 µg/mL BMP-2. The results showed greater cell proliferation in group A. The upregulation of runt-related transcription factor 2 and osteocalcin genes correlated with the release of BMP-2 from the scaffolds. The PCL-30% HA MSMD scaffolds appear to be suitable for use as osteoconductive frameworks and BMP-2 carriers.

## 1. Introduction

Severe alveolar bone loss and jawbone atrophy are common problems associated with denture and implant construction in elderly patients. These conditions significantly reduce chewing ability, which affects nutrient uptake, health status, and quality of life. At present, grafting with particulate bone grafts or bone substitute particles is the standard method for reconstructing such defects. The graft particles are thought to act as osteoconductive frameworks that support new bone regeneration in the defect areas. However, their capacity to repair large intra-bony defects is limited, especially when augmentation procedures are required. The primary problem is that the graft particles cannot maintain their volumes and shapes by themselves and so additional rigid covering membranes or mesh trays are needed. In these cases, using scaffold-formed bone substitutes could overcome this problem and increase the success rate of these treatments [1,2,3,4,5,6].

In this study, polymeric bone-substitute scaffolds were fabricated using the technique of melt stretching and multilayer deposition (MSMD) [7,8,9,10]. This technique is simple and inexpensive and does not require specialized equipment. The MSMD scaffolds have a three-dimensional (3-D) architecture with regular interconnected pore networks. By using melt blending, various types of biomaterials can be combined with polycaprolactone (PCL)-based scaffolds. In previous studies, chitosan (CS), hydroxyapatite (HA), and biphasic calcium phosphate (BCP) have been used as scaffold fillers [8,9,11]. These scaffolds had good osteoconductive properties and supported new bone formation in critical-size defects in animal models. Combining scaffolds with osteogenic growth factors may increase their osteoinductive properties. Bone morphogenetic protein-2 (BMP-2) is the most potent osteogenic growth factor, has been widely studied and is accepted for use in patients [12,13,14,15]. BMP-2 induces bone formation by stimulating chemotaxis of mesenchymal stem cells and increasing proliferation and differentiation of osteoprogenitor cells [16,17]. Although high success rates have been reported for using BMP-2 for inducing bone formation, a consensus has not been reached as to the optimum dose of BMP-2 and various concentrations have been used for in vitro experiments and clinical trials. The dosages of BMP-2 used in clinical treatments are up to 100 times the optimal therapeutic dosage in rodents [18]. In our recent study [11], PCL-30% HA MSMD scaffolds were combined with 400 µg/mL BMP-2 solution using a physisorption method. The scaffolds were used to repair critical-sized calvarial defects in rat models. It was found that the scaffolds could sustain the release of BMP-2 over the 14 days of the observation period. The cumulative released concentration of BMP-2 reached a maximum of 22.76 ± 2.15 µg/mL, which is within the range of 13–25 µg/mL for optimum bone induction in rodents [19]. In addition, BMP-2-soaked scaffolds covered with resorbable collagen membranes induced a new bone volume fraction (VF) of up to 57.93 ± 5.79% within 8 weeks. Commercial preparations of BMP-2 have concentrations of at least 1 mg/mL and commonly use collagen sponges or HA particles as carriers. These carriers suffer from disadvantages including their rapid degradation, poor mechanical strength, and inability to maintain their volumes. In addition, such high-dose BMP-2 preparations significantly increase the cost of treatment and are prohibitively expensive for most patients. Moreover, several previous studies [18,20,21,22,23] revealed unintended local and systemic side effects when the biological effects of BMP-2 extend beyond the defect sites. Local side effects include inflammation, hematoma, osteoclast-mediated bone resorption, inappropriate adipogenesis, and ectopic ossification. Systemic side effects such as paresthesia, palsy, loss of local skeletal and muscular functions, and tumorigenesis have also been reported. Therefore, several publications [24,25,26,27,28] support using lower dosages of BMP-2 not only for avoiding complications, but also for reducing the cost of the treatment. Hunziker, et al. [28] suggest that the osteogenic efficacy of BMP-2 is optimal when it is steadily delivered at extremely low concentrations. Correspondingly, some studies suggest that BMP-2 in nanogram quantities could have chemotaxis effects, while microgram amounts could stimulate bone differentiation [24,25,26,27]. Therefore, in this study, the effects of low dosages of BMP-2 on the proliferation of bone-forming cells and inducing osteogenic differentiation were comparatively assessed in vitro. The optimum dose, which achieved the best results for the osteoblasts in 2-dimensional (2-D) culture, was selected for incorporation into PCL-HA MSMD scaffolds (3-D culture). The responses of the cells in a BMP-2-treated 3-D scaffold were compared with those from the experiments in which BMP-2 was instead contained in the culture medium. Based on the results, improved applications of BMP-2 and scaffolds in clinical practice are suggested.

## 2. Materials and Methods

PCL pellets (Poly e-caprolactone, M_n_ 80,000 by GPC, Melting point 60 °C, melt index 1.00 g/10 min; and 125 °C /44 psi) were purchased from Sigma-Aldrich Corp., St. Louis, MO, USA. HA particles (particle size < 75 mm) and *E. coli*-derived rhBMP-2 (BMP-2) were supplied by the National Metal and Materials Technology Center (MTEC, Bangkok, Thailand). 

### 2.1. Scaffold Fabrication

The PCL-30% HA scaffolds were fabricated using the MSMD technique according to the following protocol [29]: the PCL pellets and the HA particles were homogeneously mixed in the ratio of PCL:HA = 70:30 by weight in a melt extrusion machine at 140 °C while stirring. Monofilaments were fabricated by extruding the mixture through the nozzle tip of the machine. Afterwards, the filaments were stretched using a universal testing machine (Lloyd, London, England) to decrease their average diameter to 0.5 mm, and they were stored in a desiccator cabinet until required for use. The monolayer scaffolds were fabricated by arranging the filaments on a polyvinyl template (3M ESPE, St Paul, MN, USA) as a grid pattern of filament lines perpendicular to each other with an average space area of 500 µm^2^. The contact points of the scaffolds were fused through compression using compressing plate and heating to 60 °C in a hot air-flow oven for 10 min. To prepare the scaffolds for the experiments, the monolayer scaffolds were cut into circles of 8 mm diameter (Figure 1A,B) and deposited in three layers before the compression step was repeated (Figure 1C). The scaffolds were kept in sterilization pouches and sterilized using ethylene oxide gas (ethylene oxide 100%, 37 °C, humidity 76%, 2 h) 2 weeks prior to the experiments. 

### 2.2. Scaffold Morphologies

The macro/microstructure of the scaffolds were analyzed using a stereomicroscope (Nikon SMZ1500, Kanagawa, Japan) and a scanning electron microscope (SEM, JEOL Ltd., Tokyo, Japan). Dispersion of the HA filler in the PCL matrix of the scaffold filaments were investigated using a micro-CT (µ-CT) machine (µCT 35, SCANCO Medical AG, Brüttisellen, Switzerland). Filament samples 30 mm in length were randomly selected and scanned in a cross-sectional direction along the length using a setting of 55 kVp, 72 mA, and 4 W. The gray scale threshold values were adjusted to highlight the HA particles in the PCL matrix.

### 2.3. Mechanical Testing

To simulate an intra-oral circumstance, the mechanical properties of the scaffolds were assessed under wet condition. The scaffolds were immersed and incubated in simulated body fluid (SBF) [30] at 37 °C for 24 h prior to the experiment. The fluid consists of ion concentrations that are equal to those of human blood plasma. Compression forces were applied to the superior aspects of the scaffolds from 0 to 200 N at a crosshead speed of 10 mm/min using a universal testing machine (Lloyd Instruments Ltd, West Sussex, UK) (n = 5) (Figure 2). The compressive strength of the scaffolds was measured using analysis software (NEXYGEN, Lloyd Instruments Ltd, Hampshire, UK).

### 2.4. Preparation of BMP-2 Solution

The BMP-2 solution was prepared by dissolving 200 µg freeze-dried BMP-2 in 200 µL of sodium acetate solution (Sigma-Aldrich Corp., St. Louis, MO, USA) at pH 4.8. The solution was sterilized through a 0.2 µm pore filter (Merck-Millipore, Darmstadt, Germany). Thereafter, the solution was stored at 4 °C and used within 24 h.

### 2.5. Finding the Optimum BMP-2 Concentration

#### Cell Culture

Osteoblasts (MC3T3-E1 cell line, subclone 4, ATCC, Manassas, VA, USA) were grown in the proliferation medium α-MEM (Gibco, Thermo Fisher Scientific, Waltham, MA, USA) supplemented with 10% fetal bovine serum (Gibco, Thermo Fisher Scientific, Waltham, MA, USA) and antibiotics (100 µg/mL penicillin G, 50 µg/mL gentamicin, and 3 mg/mL amphotericin) (Gibco, Thermo Fisher Scientific, Waltham, MA, USA). The cells in passages 3–5 were used for the experiments. An overview of the experiments is shown in Figure 3. The 2-D culture experiments were performed to evaluate proliferation and differentiation of the osteoblasts in media containing five different concentrations of BMP-2. The optimum concentration of the BMP-2 solution found in these experiments was selected for further experiments in 3-D culture.

### 2.6. Comparative Assessment of Proliferation and Differentiation of the Cells in Media Containing Five Different Concentrations of BMP (2-D Culture)

Cells (1 × 10^4^) were seeded into each well of the 48-well tissue culture plates (Nunc, Thermo Fisher Scientific, Waltham, MA, USA) and left for 3 h in 5% CO_2_ at 37 °C to allow for cell attachment. Afterwards, the medium was removed and replaced with 200 mL proliferation medium containing BMP-2 concentrations of 0.1 µg/mL (group C1), 0.5 µg/mL (group C2), 1 µg/mL (group C3), 10 µg/mL (group C4), and 50 µg/mL (group C5), and a medium without BMP-2 (control group). The plates were cultivated in 5% CO_2_ at 37 °C, and the medium was changed every 3 days until the end of the experiments. On culture days 3, 7, and 14, cell proliferation and cell differentiation assays were performed (n = 4/group/time point).

#### 2.6.1. Cell Proliferation Assay

PrestoBlue Reagent (Thermo Fisher Scientific, Waltham, MA, USA) was used to measure the viability of the cells of each group according to the following protocol: ten microliters of the PrestoBlue solution were diluted in fresh medium (1:9) and added to each well. The culture plates were incubated at 37 °C for 60 min. The optical density (OD) of each well was measured at a wavelength of 600 nm using a spectrophotometer (Thermo Fisher Scientific, Waltham, MA, USA).

#### 2.6.2. Cell Differentiation Assay

Total RNA of the cells of each group was extracted using TRIzol reagent (Invitrogen, Thermo Fisher Scientific, Waltham, MA, USA) according to the manufacturer’s protocol. Reverse Transcriptase (SuperScript III, Invitrogen, Thermo Fisher Scientific, Waltham, MA, USA) was used for reverse transcription of total RNA to cDNA. Expression levels of the osteoblast-related genes alkaline phosphatase (ALP), osteocalcin (OCN), runt-related transcription factor 2 (RUNX2), collagen type 1 (Col-1) and bone sialoprotein (BSP) were measured using Faststart SYBR Green Master kit (Sigma-Aldrich Corp., St. Louis, MO, USA) and a real-time PCR System (LightCycler 96, Roche Diagnostics International AG, Rotkreuz, Switzerland). The expression of the genes was analyzed using the 2^−ΔΔCt^ method and normalized to the expression of glyceraldehyde 3-phosphate dehydrogenase (GAPDH) housekeeping gene (Table 1).

### 2.7. Comparative Assessment of Proliferation and Differentiation of the Cells in the Cell-Scaffold Constructs (3-D Culture)

The study groups were divided into three groups. Group A consisted of PCL-30% HA MSMD scaffolds (diameter 8 mm) that were immersed in the optimum BMP-2 solution for 24 h prior to the experiment. Cells were seeded into the scaffolds at a density of 10^6^ cells/scaffold, and the cell-scaffold constructs were cultured in 500 µL of the proliferation medium. For Group B, the cells were seeded onto scaffolds without BMP-2 pre-treatment, and the constructs were cultured in 500 µL of the proliferation medium containing the optimum concentration of BMP-2. Group C was a control group. For this group, the cells were seeded on the scaffolds without prior BMP-2 treatment, and the constructs were cultured in 500 µL of the proliferation medium not containing BMP-2. The constructs of all groups were cultivated in 5% CO_2_ at 37 °C and the medium was changed every 3 days until the end of the experiments.

#### 2.7.1. Morphologies and Behaviors of Cells in the Constructs 

At days 3, 7, 14, and 21 after seeding, the cells in the constructs were fixed in 2.5% glutaraldehyde (Sigma-Aldrich Corp., St. Louis, MO, USA) for 2 h (n = 2/group/time point). They were then dehydrated in an ethanol series of 30–100%, then dried and coated with gold-palladium. The characteristics of the cells were then examined via SEM.

#### 2.7.2. BMP-2 Releasing Assay

The scaffolds were soaked in the optimum BMP-2 solution for 24 h prior to the experiment. Afterwards, the scaffolds were placed into new 48-well plates, and 500 µL of phosphate buffer saline (PBS) was added into each well. On days 1, 3, 7, 14, 21, and 28, the solution in each well was collected to quantify the amounts of BMP-2 (n = 5/timepoint), and then, fresh PBS was replaced. The amount of BMP-2 was quantified using an enzyme-linked immunosorbent assay (Quantikine, R&D Systems, Inc., Minneapolis, MN, USA) according to the manufacturer’s instructions.

#### 2.7.3. Cell Proliferation and Differentiation Assays

On culture days 3, 7, 14, and 21, the cell proliferation and differentiation assays were performed as previously described (n = 4/group/time point). For osteogenic differentiation, the expression of ALP, OCN, RUNX2, Col-1, and BSP were measured (Table 1).

### 2.8. Statistical Analysis

The microscopic features of the cell-scaffold constructs were evaluated qualitatively. The measured parameters, including the amounts of BMP-2 released from the scaffolds, the number of cells in the constructs, and the levels of the osteoblast-related genes were analyzed statistically. One-way analysis of variance (ANOVA) followed by Tukey HSD (Honestly Significant Difference) was applied to compare the differences among the groups and time points. The level of statistical significance was set at *p* < 0.05.

## 3. Results

### 3.1. Morphologies of the Scaffolds

The morphologies of the scaffolds are shown in Figure 4. The results showed that the scaffolds had a regular interconnective pore architecture. SEM images revealed rough and irregular surfaces of the scaffolds with the HA particles deposited within the PCL matrix. The µ-CT images demonstrated the dispersion of HA particles throughout the entire area of the filaments.

### 3.2. Mechanical Properties

The mechanical properties of the scaffolds are demonstrated in Table 2. The scaffolds could withstand the compression forces from the superior direction and recover to their initial height after applying the forces.

### 3.3. Responses of the Osteoblasts in 2-D Culture

#### 3.3.1. Cell Proliferation

The number of cells in all groups increased with time. On day 14, the growth of all groups remarkably increased. The overall proliferation of the cells in group C4 was greater than that in the other groups, whereas that of group C5 was far less than that of the other groups (Figure 5).

#### 3.3.2. Cell Differentiation

Expression profiles of the osteoblast-related genes are shown in Figure 6. The levels of Col-1 and BSP in group C4 were upregulated higher than those of the other groups on the first 7 days, but they were not statistically different. The levels of ALP and OCN of all groups were not statistically different over the observation period.

The results of the 2-D culture show that the optimum BMP-2 concentration to promote cell proliferation is 10 µg/mL. Therefore, this concentration of BMP-2 was used for 3-D culture experiments.

### 3.4. Responses of Osteoblasts in 3-D Culture

#### 3.4.1. Morphologies and Behaviors of Cells in the Constructs

The morphologies of the cells in the constructs are shown in Figure 7. After seeding, the cells attached well to the surfaces of the scaffolds. From day 3, the cells of all groups grew in multiple layers covering the entire surface of the scaffolds. By observation, there was no readily apparent difference in the cell behavior within each experiment group.

#### 3.4.2. Cell Proliferation

Profiles of cell proliferation are shown in Figure 8. The number of cells in groups A and C gradually increased to reach their maximum growth on day 21, whereas those of group B reached their maximum growth on day 7 and slightly decreased thereafter. The overall proliferation of the cells in group A was greater than that of the other groups.

#### 3.4.3. Cell Differentiation

Levels of the osteoblast-related genes in the experimental groups are shown in Figure 9. There was no significant difference in gene expression among the groups over the observation period. On day 3, upregulation of RUNX2 and OCN was detected in groups A and B. The expression of RUNX2 gene in group A was higher than that in the other groups until day 7.

### 3.5. Release of BMP-2 from the Scaffolds

The profiles of BMP-2 release from the scaffolds are shown in Figure 10. The rate of BMP-2 release increased to reach a maximum on day 7 and decreased thereafter. The cumulative volume of BMP-2 released on the first 7 days was 0.829 ± 0.4 µg/mL or 14.86% and the total release was 1.762 ± 0.15 µg/mL.

## 4. Discussion

This study demonstrated the efficacy of low doses of BMP-2 for enhancing growth and differentiation of bone forming cells in 2-D and 3-D culture models. Various concentrations of BMP-2 were compared in a 2-D culture to determine the optimum amount of BMP-2 for promoting cell proliferation and differentiation of osteoblasts. In previous studies, concentrations in the range of 1–50 µg/mL have been reported for their potency to promote new bone formation [31,32,33]. Mumcuoglu et al. [31] fabricated an injectable hydrogel composed of BMP-2-loaded recombinant collagen-based microspheres and alginate using doses of 50 µg/mL, 15 µg/mL, and 5 µg/mL of BMP-2. Subcutaneous ectopic bone formation and bone regeneration in calvarial defects were assessed in animal models. The result showed a higher bone volume over 8 weeks in the group of 50 µg/mL BMP-2, whereas the group with 5 µg/mL BMP-2 failed to heal the calvarial defect faster than the material without the BMP-2. The authors concluded that combining 50 µg/mL BMP-2 with the delivery system showed promising results in both ectopic and calvarial defect models. Song et al. [32] assessed the efficacy of porous PCL-BCP composite scaffolds for immobilizing collagen and 10 µg/mL of BMP-2 both in vitro and in rat models. The results demonstrated that the scaffolds combined with BMP-2 had more cell proliferation and better bone formation when compared with the scaffolds without BMP-2. Kim et al. [33] combined the PCL-20%TCP scaffolds with 50 µg/mL rhBMP2 and implanted them into the scapular bones of adult beagle dogs. The authors found that the bone in-growth in the scaffolds with BMP-2 was higher than that of the scaffolds without BMP-2. In our study, the results of the 2-D culture demonstrated that the concentration of 10 µg/mL BMP-2 enhanced proliferation and differentiation of the osteoblasts better than the other concentrations. The proliferation of the cells was higher than that of the other groups over the observation period. The expression of Col-1 and BSP genes in the group with 10 µg/mL BMP-2 were higher than those in the other groups during the early period of culture, whereas the levels of ALP and OCN were upregulated in the later phase. Interestingly, the medium containing 50 µg/mL BMP-2 failed to support cell proliferation, and the number of cells in that group was lower than that in the other groups at every time point. For the 3-D culture, two different models were used to examine the effect of different BMP-2 dosing methods. The first model was to pre-treat the scaffold by absorbing BMP-2 solution using a physisorption method. In contrast, the second model was to add BMP-2 to the culture medium and use scaffolds not pre-treated with BMP-2. The PCL-30% HA MSMD scaffolds combined with the optimum dose of 10 µg/mL BMP-2 were used for both. The experiments aimed to simulate the clinical applications of pre-incorporating BMP-2 into the carriers and intra-operative loading of BMP-2 into defect sites. It was found that the efficacy of the first model was superior to that of the second model in terms of promoting proliferation of the cells, while the cell differentiation of both groups was not significantly different. These results correspond to those of our previous study [11], which showed that the method of incorporating BMP-2 into the scaffolds prior to implanting into the cavarial bone defects better stimulates bone formation (57.93 ± 5.79%) when compared with the method of directly dripping the BMP-2 solution into the defects (43.67 ± 6.34%). Using scaffolds pre-treated with BMP-2 reduces the complexity of surgeries as it eliminates the necessity of dripping BMP-2 solution into the defects. This method does not require secondary carriers such as collagen membranes for retaining the BMP-2 in defect sites, thus reducing the cost of treatment. Additionally, using BMP-2 pre-treated scaffolds will also likely eliminate the side effects produced when the usual higher concentrations of BMP-2 leak beyond the defect sites. However, the effectiveness of BMP-2 pre-treated scaffolds should be further investigated in animal models. In this study, it was shown that PCL-30% HA MSMD scaffolds can act as an osteoconductive framework and a BMP-2 carrier. The scaffolds had acceptable mechanical properties that were comparable to those of human cancellous bone [30]. Furthermore, their compressive strength is still comparable to that of the pure-PCL MSMD scaffolds [34]. Therefore, it is presumed that the additional 30% HA filler does not affect the mechanical properties of the PCL-based scaffolds. The data also demonstrated that the scaffolds could withstand compressive forces up to 200 N; therefore, they would have adequate mechanical strength against wound contraction during the soft tissue healing process. Regarding the results of our previous study [5], the architectures of the scaffolds could maintain over the 8 weeks of the observation period, when they were placed into rat’s calvarial defects. Moreover, new bone was found regenerating throughout the interconnecting spaces of the scaffolds, therefore, the mechanical strength of the defect areas would become increasing with time during the bone healing process.

SEM and the µ-CT images demonstrated homogenous dispersion of the HA particles throughout the PCL matrix in the scaffold filaments. We presume that voids and grooves of the surfaces, with deposition of the HA particles, are the main factors for retaining the BMP-2 on the scaffolds. The voids and grooves present a high surface area for BMP-2 absorption and the exposed HA crystals would have potential electrostatic interaction with the functional groups of the BMP-2 molecules [35,36,37]. The results demonstrate that the architecture and surface topography of the scaffolds were suitable for cell attachment and proliferation. Interestingly, absorbed BMP-2 on the surfaces of the scaffolds promoted cell proliferation in the cell-scaffold constructs better than BMP-2 in the culture medium surrounding the constructs. This implies that a scaffold incorporated with BMP-2 has the potential to recruit undifferentiated cells from the local environment to attach and enhance their growth when implanted into living bone. Regarding the releasing profile of BMP, most of the BMP molecules (14.86%) were released over the first 7 days; however, the release was sustained over the whole observation period. It was found that the total released concentrations of BMP-2 from the scaffolds over 28 days were 1.762 ± 0.15 µg/mL, which were higher than the minimum concentration of 10 ng/mL for upregulating osteogenesis of mesenchymal stem cells [38]. It is possible to explain that when the binding sites of the HA molecules are occupied by the BMP-2 molecules, the surplus unbound molecules are stored physically and released early in the culture period. Yuan et al. [18] suppose that the bone formation process is not related to the early release of BMPs and the total amounts of BMPs that are applied, but it depends on functional BMP molecules, which are chemically absorbed in the calcium phosphate ceramic. Several previous studies [39,40,41,42,43] have discussed the biphasic profile of BMP-2 during the healing process of bone fracture and support that transient burst release of BMP-2 followed by sustained release is better than continuously sustained release. The burst release would correlate with the retention and induction of host cells to the scaffolds, while the sustained release would have functions to coordinate the distribution of cells inside the scaffolds. The results of the present study reveal that early release of BMP-2 in group A on the first 7 days was correlated with upregulation of *RUNX2* and *OCN* genes. Interestingly, the presence of the BMP-2, whether dissolved in the medium or absorbed onto the scaffolds, was related to the upregulation of these genes. It is known that BMP-2 increases RUNX2 expression and transactivity through BMP receptor signals [44]. RUNX2 is an essential transcription factor for the commitment of mesenchymal stem cells to the osteoblast lineage and influences osteoblast differentiation. In addition, RUNX2 is the major regulator of osteoblastic marker gene expression, including Col-1, *OPN*, *BSP*, and *OCN* [45]. Osteocalcin is the most abundant non-collagenous protein expressed in bone, specifically in cells of an osteoblast lineage, including mature osteoblasts. Jang et al. suggested that BMP-2 can activate activating transcription factor 6 (ATF6), which increases the expression of osteocalcin by directly binding to the TGACGT sequences on the osteocalcin promoter gene [44]. Nevertheless, the levels of all osteoblast-related genes among the experimental groups were slightly different. A possible explanation is that bone formation depends not only on the amount of BMP-2, but also on the bioactivity of the materials. Yuan et al. [18] suggested that the action of inducing bone formation of BMP-2 would not be dose dependent when it is combined with osteo-inductive carriers such as calcium phosphate ceramics. The authors support the proposal that amounts of BMP-2 can be reduced to the lowest dosages when using ceramic materials as carriers. Therefore, this strategy provides the possibility of clinically using BMP-2 in a more effective and safe manner.

## 5. Conclusions

The PCL-30% HA MSMD scaffolds are suitable for use as osteoconductive frameworks, and low-dose BMP-2 promotes growth and differentiation of the bone-forming cells.

## Figures and Tables

**Figure 1 polymers-13-00466-f001:**
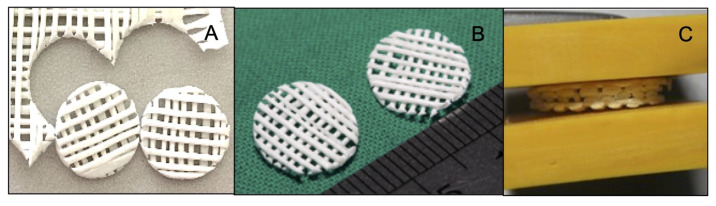
(**A**) and (**B**) The monolayer scaffolds were cut into circles. (**C**) The multilayer scaffold was made by stacking units of monolayer scaffold.

**Figure 2 polymers-13-00466-f002:**
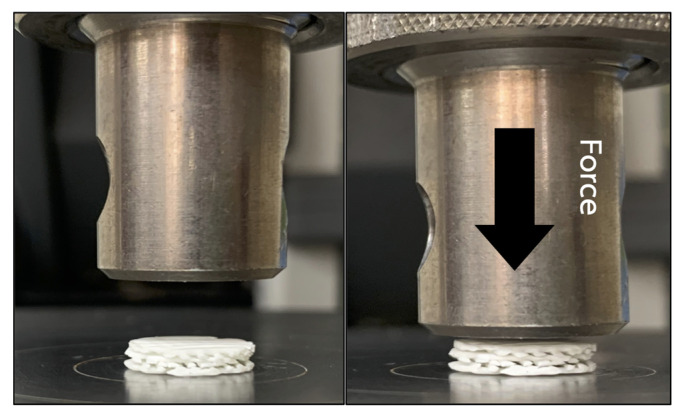
The mechanical testing was performed by applying compression force (arrow) onto the superior aspect of the scaffold.

**Figure 3 polymers-13-00466-f003:**
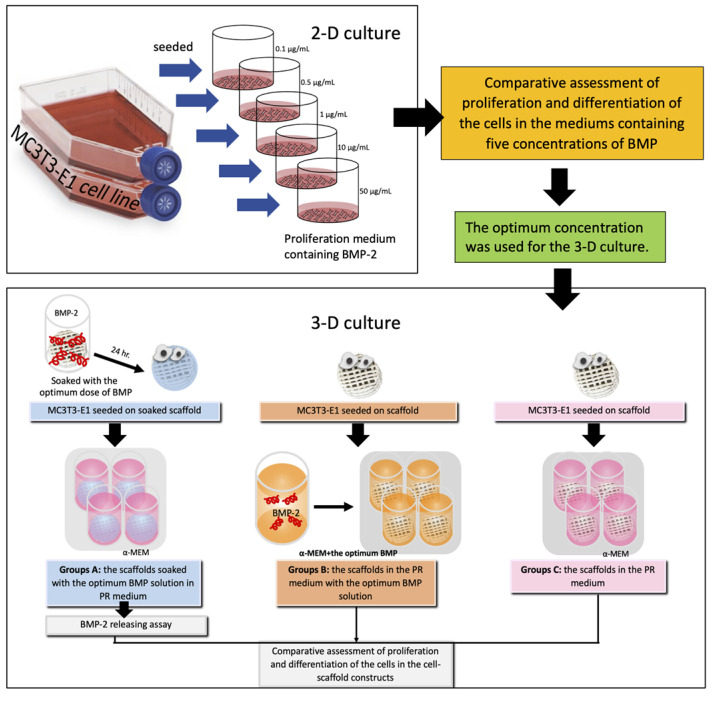
Schematic illustration of the study groups and the experiments.

**Figure 4 polymers-13-00466-f004:**
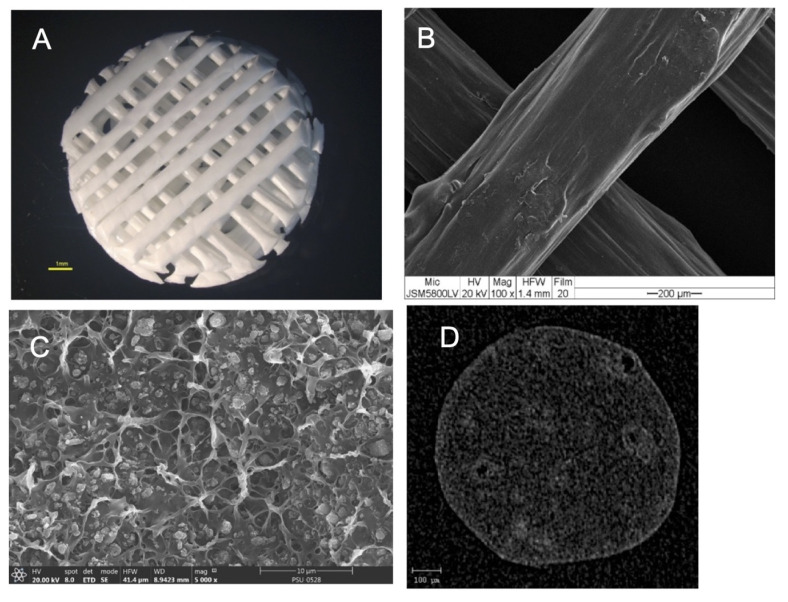
(**A**) Stereomicroscope image of the scaffold showing the regular interconnective pore architecture. (**B**) SEM image of the scaffold showing the rough and irregular surfaces. (**C**) A magnified cross-sectional SEM picture shows the HA particles deposited throughout the PCL matrix. (**D**) A cross-sectional µ-CT image shows the dispersion of the HA particles throughout the filament.

**Figure 5 polymers-13-00466-f005:**
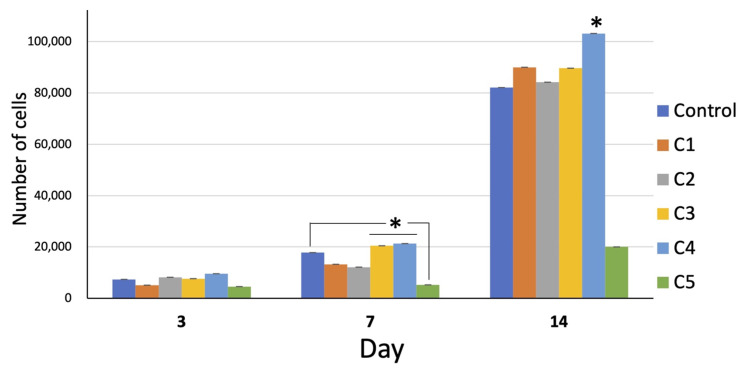
Cell proliferation of all groups over 14 days. On day 7, the proliferation of the cells of group C3 and C4 was significantly greater than that of the other groups (* = *p* < 0.05). On day 14, the proliferation of group C4 was significantly greater than the other groups (* = *p* < 0.05). Interestingly, the proliferation of group C5 was far less than that of the other groups at all time points (* = *p* < 0.05 on day 14).

**Figure 6 polymers-13-00466-f006:**
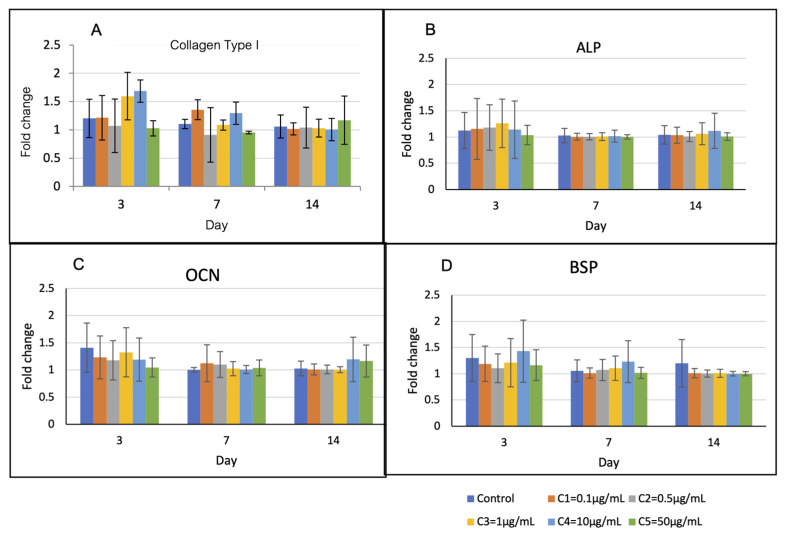
Osteoblast-related gene expression over 14 days. (**A**) On day 3, the levels of Col-1 of groups C3 and C4 were higher than those of the other groups (*p* > 0.05). (**B**) and (**C**) The levels of ALP and OCN of all groups were not statistically different over the observation period. (**D**) On day 3 and 7, the levels of BSP of group C4 were higher than those of the other groups (*p >* 0.05).

**Figure 7 polymers-13-00466-f007:**
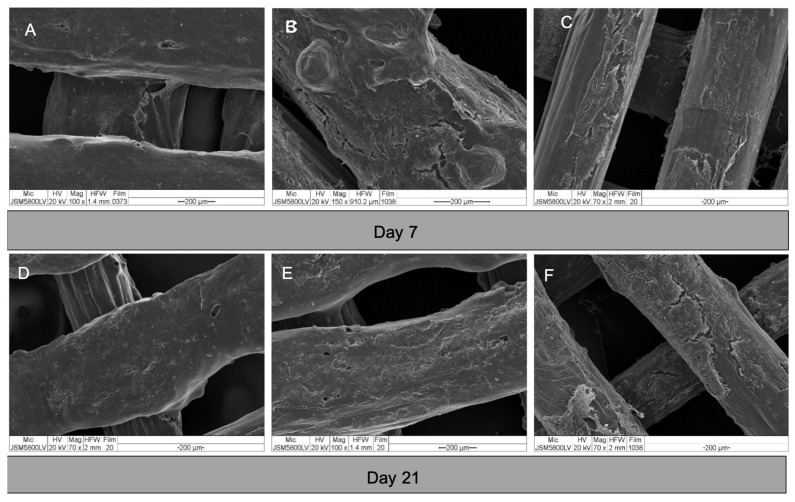
SEM images of cell-scaffold constructs on day 7 and 21. (**A**) and (**D**) group A; (**B**) and (**E**) group B; and (**C**) and (**F**) group C. The cells of all groups grew in dense multi-layers covering the entire surfaces of the scaffolds.

**Figure 8 polymers-13-00466-f008:**
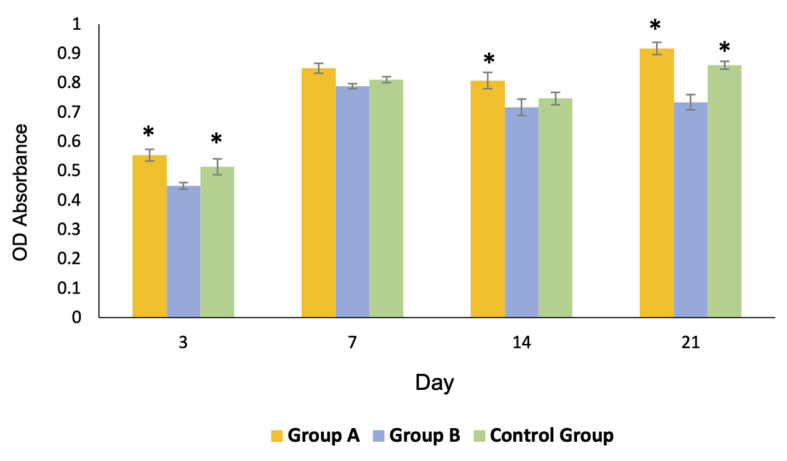
Profiles of the cell proliferation of all groups over 21 days. The growth of the cells of group A and the control group C was significantly greater than that of group B on day 3 and 21 (* = *p* < 0.05). The growth of group A was significantly greater than that of the other groups on day 14 (* = *p* < 0.05).

**Figure 9 polymers-13-00466-f009:**
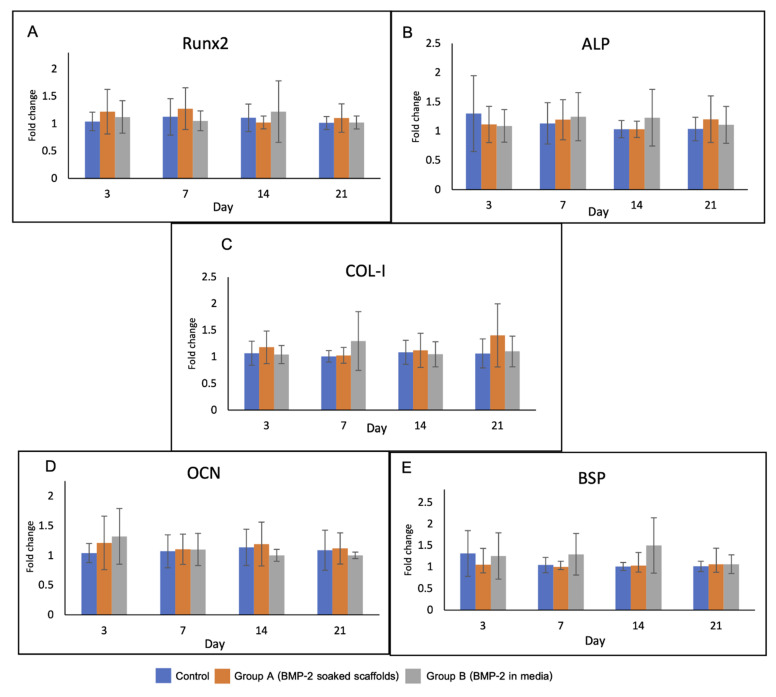
Gene expression of the cells in the scaffolds over 21 days including (**A**) RUNX2, (**B**) ALP, (**C**) Col-1, (**D**) OCN and (**E**) BSP. RUNX2 and OCN genes were upregulated in group A and B on day 3. Expression of RUNX2 in group A was higher than that in the other groups until day 7.

**Figure 10 polymers-13-00466-f010:**
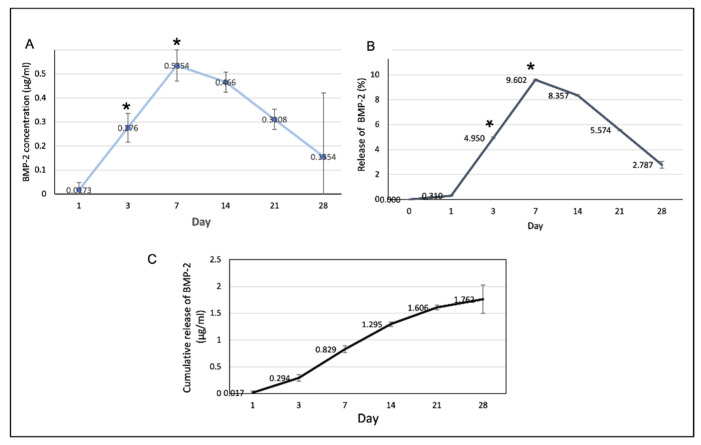
(**A**) The amount of the BMP-2 released from the scaffolds at each time point. At days 3 and 7, the release of the BMP-2 significantly increased (* *p* < 0.05). (**B**) The percentages of the BMP-2 released from the scaffolds in at each time point. (**C**) The cumulative release of the BMP-2 over the 28-day experimental period.

**Table 1 polymers-13-00466-t001:** The primer sequences.

Gene	Reverse (5′–3′)	Forward (3′–5′)
Col-1	ACCAGGTTCACCGCTGTTAC	GTGCTAAAGGTGCCCAATGGT
BSP	AGGATAAAAGTAGGCATGCTTG	ATGGCCTGTGCTTTCTCAATG
ALP	GCGGCAGACTTTGGTTTC	CCACCAGCCCGTGACAGA
RUNX2	TGCTTTGGTCTTGAAATCACA	TCTTAGAACAAATTCTGCCCTTT
OCN	CTTTGTGTCCAAGCAGGAGG	CTGAAAGCCGATGTGGTCAG
GADPH	CCACCACCCTGTTGCTGTA	GCATCCTGGGCTACACTGA

**Table 2 polymers-13-00466-t002:** The mechanical properties of the scaffolds.

Mechanical Properties	Mean ± SD
Stress at maximum load (MPa)	3.98 ± 0.007
Strain at maximum load (%)	44.33 ± 3.4
Young’s modulus (MPa)	16.22 ± 1.07

## Data Availability

The data presented in this study are available on request from the corresponding author.

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
