# Peer review of "Bioactivity of a Novel Polycaprolactone-Hydroxyapatite Scaffold Used as a Carrier of Low Dose BMP-2: An In Vitro Study"

_polymers, 2021, doi:10.3390/polym13030466_

Round 1

Reviewer 1 Report

The paper is focused on the realisation and biological validation of PCL/hydroxiapatite 3D scaffolds, dippped in BMP-2 or cultured in BMP-2 solution. 

My principal concerns are:

1) mechanical properties of the scaffolds in dry and wet conditions are not present

2) in the state of art all the work of prof. Monica Mattioli-Belmonte are not reported, she already perfomed similar experiments

3) the time dependent release of BMP-2 from the scaffold in the group A should be analysed

the paper is well written in each part.

the results should be better discussed on the basis of the previous concerns

Author Response

Reviewer 1

The paper is focused on the realization and biological validation of PCL/hydroxyapatite 3D scaffolds, dipped in BMP-2 or cultured in BMP-2 solution. 

My principal concerns are:

1) mechanical properties of the scaffolds in dry and wet conditions are not present.

Answer: The mechanical testing of the PCL-30%HA was performed in our preliminary experiments. The details of the testing and results were in the materials and methods (2.3) and in the results on page 3, line 116, and page 7, line 225. The testing was performed in wet condition for simulating situations in oral environment. The results were discussed in the discussion part on page 12, line 347-358. We found that the scaffolds had acceptable mechanical properties and would have adequate mechanical strength against wound contraction. Therefore, they are suitable for using in two and three-wall intra-bony defects.

2) in the state of art all the work of prof. Monica Mattioli-Belmonte are not reported, she already perfomed similar experiments.

Answer: Some references from Prof. Monica Mattioli-Belmonte, et al. which relate to the study were added in the introduction part on page 1, line 33.

3) the time dependent release of BMP-2 from the scaffold in the group A should be analyzed.

Answer: We added new data of the percentage of the release of BMP on the first 7 days on page 10, line 286 and the new graph and its description on page 11; Figure 10 (B) and line 295, showing the percentages of the released BMP from the scaffolds relating to the time. Therefore, the amounts of the BMP in percentages can be demonstrated. This data elucidates the characteristic and ability of the scaffolds for releasing BMP. We also added more points of discussion about the release of BMP on page 12, Line 370-377. 

- the paper is well written in each part.

- the results should be better discussed on the basis of the previous concerns.

Reviewer 2 Report

1、Page 6 line 211 , There is a lack of data after 14 days of cell proliferation, so it is not rigorous to say that the number of cells in all groups increased to reach their maximum growth on day 14. 2、Why use 2-D culture to optimize the BMP-2 concentration, instead of directly using 3D models to optimum BMP-2 concentration to promote cell proliferation ? 3、Page 7,line222 and Page 9,line254: These conclusions are not valid. The data in Figure 5 and Figure 8 have large errors and significant differences, and the effect is statistically consistent.

Author Response

Reviewer 2

Comments and Suggestions for Authors

1、Page 6 line 211, There is a lack of data after 14 days of cell proliferation, so it is not rigorous to say that the number of cells in all groups increased to reach their maximum growth on day 14.

Answer: We agree with your comment that day 14 is not the day of maximum cell growth. Therefore, the result on page 7, line 233 is changed to be “The number of cells in all groups increased with time. On day 14, the growth of all groups remarkably increased.”.

2、Why use 2-D culture to optimize the BMP-2 concentration, instead of directly using 3D models to optimum BMP-2 concentration to promote cell proliferation?

Answer: The experiments of the 2-D culture were performed to find out the optimum concentration of the BMP solution for combining with the scaffolds in the following experiments. The reasons for varying the dosages of BMP in the 2-D culture are as follows.

  1. The initial amounts of the cells among the groups could be better controlled when they were seeded onto the flat tissue culture plates. The MSMD scaffolds have large average pore size of 500 µm, therefore, some seeded cells possibly fail to attach on the scaffold after seeding. Regarding the preliminary experiment, the amounts of cell loss during seeding are 18.77±3.05 %. The equal initial cell number of all groups is the important factor for the accuracy of the results of the cell proliferation and differentiation and avoiding misled conclusion.
  2. Accuracy of the concentrations of the BMP can be better controlled when added into the culture medium rather than absorbing into the scaffold especially when the low concentrations are experimented. This method can make the results of the cellular responses more precise to evaluate for the optimum dose.

3、Page 7, line 222 and Page 9, line 254: These conclusions are not valid. The data in Figure 5 and Figure 8 have large errors and significant differences, and the effect is statistically consistent.

Answer: We agree with your comment that the statistical differences of the gene expression must be re-evaluated. After re-analyzed the raw data, some outliers were excluded, we found that there was no statistical difference among the parameters of all groups. Therefore, the results of the 2-D and 3-D culture on page 8, line 246 and 252, and page 10, line 281 were revised. 

Reviewer 3 Report

Dear authors,

I recommend your article to be accepted for publication after minor revision.

row 188: Insert "." at the end of the sentence.

rows 280 and 287: Change "BMP2" to "BMP-2".

row 357: Insert "activating" between "activate" and "transcription" so that ATF6 fullname will be correct.

Author Response

Reviewer 3

Comments and Suggestions for Authors

Dear authors,

I recommend your article to be accepted for publication after minor revision.

row 188: Insert "." at the end of the sentence.

rows 280 and 287: Change "BMP2" to "BMP-2".

row 357: Insert "activating" between "activate" and "transcription" so that ATF6 full name will be correct.

Answer: All points of your comments were corrected (page 6 line 204, page 11 line 306 and 313, and page 13 line 395).

Round 2

Reviewer 1 Report

the paper was revised as required, so it can be published

Reviewer 2 Report

The authors have addressed all the comments from the reviewers.